# Unveiling the Connection: Viral Infections and Genes in dNTP Metabolism

**DOI:** 10.3390/v16091412

**Published:** 2024-09-03

**Authors:** Shih-Yen Lo, Meng-Jiun Lai, Chee-Hing Yang, Hui-Chun Li

**Affiliations:** 1Department of Laboratory Medicine and Biotechnology, Tzu Chi University, Hualien 970, Taiwan; losylo@mail.tcu.edu.tw (S.-Y.L.); monjou@mail.tcu.edu.tw (M.-J.L.); cheehing2@gms.tcu.edu.tw (C.-H.Y.); 2Department of Laboratory Medicine, Buddhist Tzu Chi General Hospital, Hualien 970, Taiwan; 3Department of Microbiology and Immunology, School of Medicine, Tzu Chi University, Hualien 970, Taiwan; 4Department of Biochemistry, School of Medicine, Tzu Chi University, Hualien 970, Taiwan

**Keywords:** deoxynucleoside triphosphates, dihydrofolate reductase, ribonucleotide reductase, SAMHD1, viruses

## Abstract

Deoxynucleoside triphosphates (dNTPs) are crucial for the replication and maintenance of genomic information within cells. The balance of the dNTP pool involves several cellular enzymes, including dihydrofolate reductase (DHFR), ribonucleotide reductase (RNR), and SAM and HD domain-containing protein 1 (SAMHD1), among others. DHFR is vital for the de novo synthesis of purines and deoxythymidine monophosphate, which are necessary for DNA synthesis. SAMHD1, a ubiquitously expressed deoxynucleotide triphosphohydrolase, converts dNTPs into deoxynucleosides and inorganic triphosphates. This process counteracts the de novo dNTP synthesis primarily carried out by RNR and cellular deoxynucleoside kinases, which are most active during the S phase of the cell cycle. The intracellular levels of dNTPs can influence various viral infections. This review provides a concise summary of the interactions between different viruses and the genes involved in dNTP metabolism.

## 1. Introduction

Deoxynucleoside triphosphates (dNTPs) are essential for the replication and maintenance of genomic information in cells and various viral pathogens. The metabolism of dNTPs involves several cellular enzymes, including dihydrofolate reductase (DHFR), ribonucleotide reductase (RNR), SAM domain and HD domain-containing protein 1 (SAMHD1), and thymidine kinase (TK), among others (Figure 1). Steady-state intracellular dNTP pools are maintained through regulated cellular processes dedicated to the synthesis and degradation of dNTP molecules. Enzymes like RNR and TK contribute to dNTP biosynthesis, while SAMHD1 degrades dNTPs into 2′-deoxynucleosides (dNs) and triphosphates (PPPi) through its dNTP triphosphohydrolase (dNTPase) activity [1,2,3,4].

DHFR, a key enzyme in the biosynthesis of amino acid and folic acid, is ubiquitously expressed in both prokaryotes and eukaryotes. The human DHFR sequence consists of 187 amino acids and catalyzes the regeneration of tetrahydrofolate from dihydrofolate, using NADPH as a cofactor (Figure 2). DHFR is crucial for cell growth and proliferation, as it is essential for the de novo synthesis of purines and deoxythymidine monophosphate, which are necessary for DNA precursor synthesis. This enzyme is a key target for anticancer drugs like methotrexate (MTX), which competitively inhibits human DHFR, leading to reduced thymidylate levels and impaired DNA synthesis in rapidly dividing cancer cells [5,6,7,8].

RNRs catalyze the reduction of the hydroxyl group on the second carbon of ribose, converting ribonucleotides (NTPs) into their corresponding deoxyribonucleotides (dNTPs) to promote DNA synthesis (Figure 3) [2,9]. RNR is utilized by all free-living organisms and is a highly regulated, cell cycle-controlled enzyme critical for DNA synthesis and repair [10]. Mammalian RNRs are tetramers (α2β2), consisting of two α subunits (RRM1) and two β subunits (RRM2 or p53R2). The human RRM1 sequence is 792 amino acids in length, RRM2 is 389 amino acids, and p53R2 is 351 amino acids [11]. RRM1 binds nucleoside triphosphate allosteric effectors, while RRM2 contains a tyrosine free radical essential for activity. RRM2 expression is specific to the S phase and is required for DNA replication [12], while p53R2 is expressed throughout the cell cycle, providing dNTPs for mitochondrial DNA synthesis in quiescent cells [13]. RRM1 contains the catalytic site and two different allosteric regulatory sites; RRM2 has a di-iron cofactor and a tyrosyl radical essential for RNR activity.

SAMHD1 is ubiquitously expressed and consists of 626 amino acids. This 65-kDa protein has three structural domains: (1) an N-terminal SAM domain (amino acids 45–110) preceded by a nuclear localization sequence (amino acids 11–14), (2) a central, catalytic HD domain (amino acids 164–316) with conserved metal-coordinating histidine and aspartic acid residues essential for dNTPase function, and (3) a C-terminal regulatory domain [14]. SAMHD1 maintains homeostatic dNTP levels, regulating DNA replication and damage repair. It catalyzes the hydrolysis of dNTPs into dNs and PPPi (Figure 4), acting as a negative regulator of dNTP pools [15]. This counteracts the de novo dNTP biosynthesis primarily conducted by RNR and deoxynucleoside kinases, which are active during the S phase of the cell cycle. RNR catalyzes the formation of dNTPs from ribonucleotides, while deoxynucleoside kinases, like TK, add phosphates to nucleosides to form dNTPs [3]. This ensures a balanced dNTP pool during the cell cycle, providing sufficient dNTPs for genome replication in dividing cells and limited levels in nondividing cells. The cellular enzymes involved in dNTP biosynthesis are upregulated during the S phase to support DNA replication. In contrast, the cellular SAMHD1 expression remains consistent throughout the cell cycle, accumulating in cultures upon starvation-induced quiescence. SAMHD1’s dNTPase activity is regulated through post-translationally modification, e.g., phosphorylation [3,14,16].

The regulation of intracellular dNTP pools is critical to genomic stability, cancer development, and viral infections. Imbalanced dNTP pools can lead to increased mutagenesis and cell proliferation, resulting in cancer. The interplay between dNTP metabolism and viral infections is complex [16]. Viruses rely on the host cell’s dNTPs for their replication, and the availability of these molecules can significantly affect viral replication efficiency. Understanding how different viruses interact with the host genes involved in dNTP metabolism can provide insights into viral pathogenesis and potential therapeutic targets. This review summarizes the interactions between different viruses and the genes regulating the intracellular dNTP pool, including DHFR, RNR, and SAMHD1.

## 2. Interactions between Various DNA Viruses and Genes Involved in dNTP Metabolism

Most of the common DNA viruses contain non-segmented, double-stranded DNA genomes, such as herpesviruses, poxviruses, adenoviruses, papillomaviruses, and polyomaviruses, while parvoviruses have non-segmented, single-stranded DNA genomes. These DNA viruses replicate their genomic DNAs directly from the template DNAs but not through an RNA intermediate [https://ictv.global/ accessed on 1 August 2024]. These DNA viruses depend on cellular dNTPs for the replication of their genomes. An increase in intracellular dNTPs enhances DNA virus replication. Consequently, different DNA viruses employ various strategies to upregulate cellular DHFR and/or RNR activities or downregulate cellular SAMHD1 activity, especially in non-dividing cells [17]. Large DNA viruses, such as herpesviruses and poxviruses, can enhance their replication by expressing viral DHFRs and/or viral RNRs or inducing cellular DHFRs and/or cellular RNRs. On the other hand, SAMHD1 is a cellular triphosphohydrolase that degrades intracellular dNTPs, restricting viral DNA synthesis and preventing the replication of various DNA viruses in non-dividing cells [16]. DNA viruses may suppress cellular SAMHD1 activity to enhance viral replication.

### 2.1. Herpesviruses

The herpesvirus family consists of three subfamilies: alphaherpesviruses (e.g., Herpes simplex virus type 1 (HSV-1), Herpes simplex virus type 2 (HSV-2), Varicella-Zoster virus (VZV)), betaherpesviruses (e.g., human cytomegalovirus (HCMV), human herpesviruses 6 and 7 (HHV-6, HHV-7)), and gammaherpesviruses (e.g., Epstein–Barr virus (EBV), Kaposi sarcoma-associated herpesvirus (human herpesvirus 8 (HHV-8)) [https://ictv.global/ accessed on 1 August 2024].

#### 2.1.1. DHFR

The HHV-8 genome contains a DHFR gene encoding the enzyme required for nucleotide and methionine biosynthesis [18]. In peripheral blood mononuclear cells (PBMCs) where cellular endogenous DHFR is undetectable, HHV-8 increases total DHFR activity in infected PBMCs. Viral DHFR sequences are also found in herpesvirus saimiri and herpesvirus ateles but not in EBV, Marek’s disease virus, HSV, VZV, herpesvirus tamarinus, or HCMV [19,20]. HCMV productively infects quiescent fibroblasts with low dNTP levels. To replicate efficiently, HCMV induces the expression of cellular enzymes involved in DNA precursor biosynthesis. HCMV activates the cellular DHFR promoter via the E2F site through its major immediate-early gene region (IE1 and IE2) [17,21]. Murine cytomegalovirus (MCMV) also induces cellular DHFR in quiescent cells [22]. Methotrexate (MTX), an inhibitor of DHFR, suppresses MCMV replication at the DNA synthesis step in quiescent NIH 3T3 cells [23].

#### 2.1.2. RNRs

Alphaherpesviral RNR consists of large (pUL39, RRM1) and small (pUL40, RRM2) subunits. The HSV RNR enzyme, a viral early gene product, is essential for productive acute and latent infections in mice and replication in mouse cells at 38 °C. Inhibitors of viral RNRs suppress viral replication [24,25]. Two HSV mutants defective in viral RNR activity exhibit hypersensitivity to antiviral agents like acyclovir [26]. The RNR of HSV-1 and HSV-2 is inhibited by ATP and MgCl_2_ but weakly by the ATP X Mg complex, distinguishing these virally induced enzymes from other RNRs [27,28,29]. The HSV-2 RRM1 protein kinase domain is required for viral immediate-early gene transcription, likely involved in latency reactivation [30,31]. VZV open reading frames 18 and 19 (ORF18 and ORF19) encode RRM2 and RRM1 homologues, respectively. VZV RNR is weakly inhibited by 2′-deoxynucleoside triphosphates, similar to HSV-1 RNR. The RRM1 of VZV is not essential for virus infection in vitro but its deletion impairs VZV growth in cell culture and increases susceptibility to acyclovir [32]. Pseudorabies virus, an alphaherpesvirus of swine, induces viral RNR (pUL39 and pUL40) [33,34]. Marek’s disease virus (MDV) encodes viral RNR with large (UL39, RRM1) and small (UL40, RRM2) subunits, suggesting RNR’s crucial role in DNA synthesis. HCMV encodes inactive RRM1 homologs (UL48 and UL45), relying on cellular RNR for dNTP synthesis. RNR inhibitors (e.g., hydroxyurea, didox, and trimidox) can inhibit HCMV replication [35]. EBV ORFs encode RRM1 and RRM2 homologs, while HHV-8 genes ORF61 and ORF60 encode RRM1 and RRM2, respectively [36,37]. Cyprinid herpesvirus 2 (CyHV-2), causing herpesviral hematopoietic necrosis in goldfish, uses ORF23 and ORF141 as viral RNR, essential for DNA virus replication [38].

#### 2.1.3. SAMHD1

SAMHD1 inhibits HSV-1 infection by limiting intracellular dNTP levels through its dNTPase activity in non-dividing myeloid cell lines [39]. SAMHD1 restricts HCMV replication in primary monocyte-derived macrophages (MDM) and inhibits HCMV IE gene expression by directly binding to NFκB [40]. In SAMHD1 knockout (KO) mice, MCMV replication is enhanced, indicating SAMHD1’s role as an antiviral restriction factor against MCMV. SAMHD1 depletion in latently EBV-infected Akata cells increases viral particle accumulation upon EBV reactivation, suggesting SAMHD1’s inhibition of EBV replication [41]. SAMHD1 antiviral activity is phosphorylation-dependent and regulated by cyclin-dependent kinases (CDK), controlling cell cycle and proliferation. CDKs, particularly CDK6 and CDK2, mediate SAMHD1 antiviral activity against HSV-1 [42]. DNA viruses antagonize SAMHD1 to ensure successful infection. Herpesviruses encode conserved protein kinases resembling cellular CDKs, such as UL13 of HSV-1/2, ORF47 of VZV, UL97 of HCMV, U69 of HHV-6/7, BGLF4 of EBV, and ORF36 of HHV-8. EBV BGLF4 phosphorylates SAMHD1, a common feature shared by beta- and gamma-herpesviruses, inhibiting its dNTPase activity [43,44]. In addition to phosphorylation, downregulation or relocalization of SAMHD1 are alternative evasion strategies for HCMV infection [16,45,46].

### 2.2. Poxvirus and African Swine Fever Virus

Vaccinia virus (VV) is the prototype poxvirus, notable for its structure and replication mechanisms [47]. African swine fever (ASF) is a fatal infectious disease affecting swine, caused by the African swine fever virus (ASFV) [48]. Both VV and ASFV replicate in the cytoplasm, featuring large genomes and complex structures.

#### 2.2.1. RNR

Vaccinia virus encodes both subunits of RNR, which is a viral early gene product [49]. The VV RRM1 protein is approximately 87 kDa, and the VV RRM2 protein is approximately 34 kDa [50,51]. VV RNR shares several similarities with other eukaryotic RNRs but is distinct from cellular reductase in response to certain modulators of reductase activity. It requires an activator, exogenous reducing equivalents, and iron for full activity [52,53]. Inhibitors such as hydroxyurea, EDTA, dATP, and dTTP block CDP reduction, differentiating it from herpesvirus RNR, which requires no activation and is insensitive to dNTP inhibition. As expected, hydroxyurea, an RNR inhibitor, blocks VV replication [54,55].

For ASFV, a 5.5-kb fragment from the Malawi strain was identified as containing the genes for both RRM1 and RRM2. Triapine can inhibit ASFV infection in a dose-dependent manner. Molecular docking studies indicate that triapine might interact with the active center Fe^2+^ in ASFV RRM2, suggesting that viral RNR is crucial for ASFV replication [56].

#### 2.2.2. SAMHD1

SAMHD1 significantly inhibits VV infection by limiting intracellular dNTP levels in non-dividing myeloid cell lines [39]. Infection of VV Ankara in human dendritic cells is enhanced when SAMHD1 is degraded by simian immunodeficiency virus (SIV) viral protein X (Vpx), restoring VV late gene expression [57].

Transcriptome analysis shows significant changes in SAMHD1 gene expression in pigs infected with non-lethal ASFV, implying a potential role for SAMHD1 in ASFV replication [58].

### 2.3. Adenoviruses

#### 2.3.1. DHFR

The adenovirus immediate-early protein E1A activates the adenovirus E2 promoter and several cellular gene promoters, including DHFR, through the transcription factor E2F [59]. Adenovirus infection modulates cellular DHFR gene expression by initially increasing and then decreasing the rate at which DHFR-specific mRNA sequences appear in the cytoplasm and enter the pool of mRNA available for translation. Therefore, post-transcriptional nuclear events are also important in the regulation of cellular DHFR gene expression by adenovirus [60].

#### 2.3.2. RNRs

The plant amino acid mimosine has been reported to inhibit adenovirus DNA synthesis in cultured cells by inhibiting cellular RNR [61]. This inhibition occurs because mimosine chelates the iron required for RRM2 function, and its inhibition of viral DNA synthesis is reversible by adding iron to the medium. This suggests that cellular RNR activity plays an important role in adenovirus replication.

### 2.4. Human Papillomavirus (HPV) and Polyomavirus

#### 2.4.1. RNRs

Productive replication of HPV is restricted to the uppermost layers of the differentiating epithelia. HPV31-positive cells exhibit increased dNTP pools and levels of cellular RRM2. RRM2 depletion blocks productive replication, suggesting that cellular RRM2 provides dNTPs for viral DNA synthesis in differentiating cells [62]. HPV31 regulates cellular RRM2 levels through the expression of the viral E7 protein and activation of the ATR-Chk1-E2F1 DNA damage response, which is essential to combat replication stress upon entry into the S-phase and for productive replication [62]. Thus, RRM2, as a downstream target for HPV E7, is upregulated at the transcriptional level through the E7–pRb interaction and binding of E2F to the cellular RRM2 promoter region [63].

Hydroxyurea, an inhibitor of the cellular RNR enzyme, inhibits polyoma DNA synthesis [64]. This suggests that cellular RNR activity plays an important role in polyoma replication. Similarly, mimosine, an inhibitor of RRM2, has been reported to inhibit SV40 DNA synthesis in cultured cells, indicating that cellular RNR activity is crucial for SV40 replication [61].

#### 2.4.2. SAMHD1

Infection of HPVs is responsible for the development of around 5% of all human cancers. HPV16 replication is enhanced in the absence of cellular SAMHD1, leading to the hyperproliferation of keratinocytes [65]. This suggests that cellular SAMHD1 is a restriction factor for HPV16. On the other hand, HPV16 replication was also found to convert cellular SAMHD1 into a homologous recombination factor and promote its recruitment to replicating viral DNA as a pro-viral role [66].

### 2.5. Parvovirus

#### RNRs

Mimosine, an inhibitor of cellular RRM2, has been reported to inhibit parvovirus DNA synthesis in cultured cells [61]. This suggests that cellular RNR activity plays an important role in parvovirus replication.

## 3. Interactions between Various Viruses Whose Replication Is through Reverse Transcription and Genes Involved in dNTP Metabolism

Retroviruses are single-stranded, positive-sense RNA viruses while hepatitis B virus (HBV) is a DNA virus [https://ictv.global/ accessed on 1 August 2024]. Replication of retroviral RNA genomes is through retroviral DNA as an intermediate form while HBVs replicate their viral genomic DNAs through an RNA intermediate. Replication of both retroviruses and HBV requires reverse transcriptase activities. Thus, the replication kinetics of retroviruses inherently depend on the availability of cellular dNTPs for viral DNA synthesis. In rapidly dividing cells (e.g., activated CD4+ T cells), the concentrations of dNTPs are high, allowing reverse transcription to occur efficiently. In contrast, non-dividing cells (e.g., macrophages in the case of human immunodeficiency virus (HIV)) have lower dNTP pools, which restrict efficient reverse transcription. Thus, HIV and HBV may utilize various strategies to increase the dNTP pool in cells through upregulation of cellular DHFR and/or cellular RNR activities or downregulation of cellular SAMHD1 activity, which has been shown to restrict retroviruses by decreasing the pool of intra-cellular dNTPs.

### 3.1. Retrovirus

#### 3.1.1. RNRs

The completion of the life cycle of any viral infection depends on various cellular factors [67]. Macrophages are one of the major target cells for HIV-1. The CDK inhibitor p21 inhibits the replication of HIV-1 and other primate lentiviruses in human monocyte-derived macrophages (MDMs) by impairing the reverse transcription of the viral genome. The replication of retroviral genomes (e.g., HIV) during reverse transcription requires cellular RNRs to convert rNTPs to dNTPs, which are then used as substrates for DNA synthesis. Mechanistically, p21 reduces dNTP synthesis by downregulating RRM2 transcription through the repression of E2F, its transcriptional activator [68].

Treatment of HIV infection continues to be a challenge. Currently, the first-choice regimen of anti-retroviral therapy is two nucleoside reverse transcriptase inhibitors, and a protease inhibitor, non-nucleoside reverse transcriptase inhibitor, or integrase inhibitor [69,70]. Several RNR inhibitors (e.g., hydroxyurea, gemcitabine, resveratrol, clofarabine) have been reported to enhance the anti-HIV-1 activities of various nucleoside analogs (e.g., 5-azacytidine), including those that act as chain terminators and those that increase the HIV-1 mutation rate [71,72,73,74,75,76].

#### 3.1.2. SAMHD1

In CD4+ myeloid lineage cells and resting T-cells, the cellular SAMHD1 protein blocks HIV-1 infections by depleting the dNTP pool. It catalyzes the hydrolysis of dNTPs into dNs and PPPi, reducing the dNTP levels to a point that cannot support HIV-1 replication [15]. The SAMHD1 activity is inhibited by CDK-mediated phosphorylation. Specifically, CDK2 and CDK6 have been identified as mediators of SAMHD1 activation [77]. Accordingly, palbociclib, a selective inhibitor of CDK 4/6, inhibits HIV-1 reverse transcription by modulating SAMHD1 activity [78]. Similarly, the CDK inhibitor p21 has been reported to mediate HIV-1 restriction in CD4+ cells through its effect on SAMHD1 [68,79]. Thus, the cellular SAMHD1 may play a significant role in controlling virus replication in individuals with HIV and slowing disease progression [80,81].

MDMs actively express dNTP biosynthesis enzymes such as RNR, TK, and nucleoside-diphosphate kinase. However, HIV-1 replication in primary MDMs is kinetically restricted at the reverse transcription step due to the low dNTP pools established by cellular SAMHD1 [82]. In addition to HIV-1, cellular SAMHD1 has been shown to inhibit other lentiviruses, including HIV-2, several simian immunodeficiency viruses (SIV), feline immunodeficiency virus (FIV), bovine immunodeficiency virus (BIV), and equine infectious anemia virus (EIAV) [16,83]. While both HIV-1 and HIV-2 encode the accessory viral protein R (Vpr), only HIV-2 and related SIVs contain an additional open reading frame for viral protein X (Vpx) [84]. Exogenous Vpx effectively counteracts cellular SAMHD1, relieving the kinetic block on the dNTP pool, thus facilitating HIV-1 reverse transcription in MDMs [85]. Indeed, HIV-2 and several SIV strains antagonize the SAMHD1 restriction using Vpx and elevate intracellular dNTP pools to benefit their replication [16,86]. Some other SIV strains without Vpx could antagonize the cellular SAMHD1 restriction through Vpr [84].

Another strategy to evade the dNTPase activity of SAMHD1, found in HIV-1 and other SIVs lacking Vpx, is the use of highly efficient reverse transcriptase molecules that function at low dNTP concentrations in the presence of cellular SAMHD1 [87,88]. SAMHD1-mediated restriction does not affect foamy viruses, which enter new target cells with an almost completely reverse-transcribed viral genome [89].

Nucleoside reverse transcriptase inhibitors (NRTIs), commonly used antiretroviral agents, compete with intracellular dNTPs as substrates for viral reverse transcriptase. Consequently, cellular SAMHD1 activity may influence the efficacy of NRTIs in inhibiting viral replication. Indeed, it has been reported that reducing SAMHD1 levels significantly decreases HIV sensitivity to thymidine, but not to other nucleotide analogs, in both macrophages and lymphocytes [90].

### 3.2. Hepatitis B Virus (HBV)

HBV is one of the smallest human pathogens, with a DNA genome of only 3.2 kb. It successfully completes its life cycle by hijacking the cellular machinery [91]. HBV infects non-dividing liver cells, where dNTP levels are limited. However, dNTPs are essential for efficient HBV replication. Therefore, HBV likely employs various strategies to increase the dNTP pool within infected cells.

#### 3.2.1. RNRs

HBV infects and replicates in quiescent hepatocytes, which are deficient in the dNTPs crucial for the reverse transcription step of HBV replication. Unlike other tumor viruses that increase dNTP biosynthesis by inducing cell proliferation, HBV acquires dNTPs by activating the expression of cellular RRM2, a cell cycle gene that is rate-limiting for dNTP production, without inducing the cell cycle [92]. Mechanistically, HBV enhances RRM2 expression by activating Chk1, a known E2F kinase that responds to DNA damage [93]. In cells where Chk1 was pharmacologically inhibited or depleted by shRNA-mediated knockdown, HBV-mediated RRM2 expression was significantly reduced [94]. Additionally, HBV has been found to attenuate DNA repair, thereby reducing cellular dNTP consumption [93].

Further studies have shown that the HBV X protein (HBx) is linked to the elevation of cellular RRM2 expression [95,96]. A small conserved region of 125 bases within the HBx gene, named the ERE, is sufficient to upregulate RRM2 expression in growth-arrested HepG2 cells and primary human hepatocytes [97]. Interestingly, there is a high sequence similarity between ERE and a region within the RRM2 5′ UTR, named the R2-box. The ERE might directly target the RRM2 gene via the R2-box. Serum RRM2 level has been reported to be a reliable biomarker for accurately diagnosing and evaluating HBV-related cirrhosis. It also reflects the expression state of HBV replication in patients with HBV-related cirrhosis [98].

As expected, inhibitors of cellular RNR activity, such as hydroxyurea, pterostilbene, and iron depletion with desferrioxamine, lead to a significant reduction in HBV production. Additionally, osalmid and its derivative 4-cyclopropyl-2-fluoro-N-(4-hydroxyphenyl) benzamide significantly inhibited HBV DNA and cccDNA synthesis in HepG2.2.15 cells by targeting RRM2 [99]. These results suggest that cellular RRM2 may be a promising target for HBV inhibition.

#### 3.2.2. SAMHD1

Sufficient dNTP molecules are essential for efficient HBV replication. Thus, silencing cellular SAMHD1 in hepatic cells increases HBV replication due to higher dNTP levels, while overexpression has the opposite effect [100]. SAMHD1 significantly affects the levels of extracellular viral DNA and intracellular reverse transcription products. Mutations in SAMHD1 that interfere with its dNTPase activity (D137N), the catalytic center of the histidine-aspartate (HD) domain (D311A), and a phospho-mimetic mutation (T592E) abrogate its inhibitory activity. Therefore, SAMHD1 has been identified as a factor that reduces the infectivity of HBV by restricting reverse transcription through reducing intracellular dNTP levels [101]. Recently, a dual role for SAMHD1 in HBV replication has been reported: facilitating cccDNA genesis while restricting reverse transcriptase-dependent particle genesis [102]. SAMHD1 is known to be regulated by CDK-mediated phosphorylation. Among various CDK members, CDK2 has been reported to play a greater role in regulating SAMHD1 phosphorylation and HBV replication than CDK1 or CDK6 [103].

## 4. Interactions between Various RNA Viruses and Genes Involved in dNTP Metabolism

Most of the common RNA viruses contain non-segmented, single-stranded, positive-sense RNA genomes, such as coronaviruses, Zika virus, hepatitis C Virus (HCV), Chikungunya virus, dengue virus, picornaviruses, togaviruses, and Hepevirus, while respiratory syncytial virus, paramyxoviruses, filoviruses, and rhabdoviruses have non-segmented, single-stranded, negative-sense RNA genomes. Influenza A virus contains eight-segmented, single-stranded, negative-sense RNA genomes, while reoviruses have segmented, double-stranded RNA genomes [https://ictv.global/ accessed on 1 August 2024]. The relationship between cellular NTP levels and RNA viruses (excluding retroviruses, which replicate through a DNA intermediate) has not been extensively studied. It is reasonable to assume that higher intracellular NTP levels could enhance the replication of RNA viruses. Thus, different RNA viruses may employ various strategies to increase the NTP pool in cells. However, the concentration of NTPs in mammalian cells is several orders of magnitude higher than that of dNTPs [104,105]. Consequently, the NTP pool may already be sufficient for the replication of RNA viruses under most conditions, making the promotion of NTP levels during RNA virus infection potentially unnecessary.

### 4.1. Coronaviruses

#### DHFR

Methotrexate (MTX), a cellular DHFR inhibitor, can suppress the entry and replication of severe acute respiratory syndrome coronavirus 2 (SARS-CoV-2), the virus responsible for COVID-19, by targeting furin and the host DHFR enzyme, respectively. Additionally, MTX has been shown to inhibit all four SARS-CoV-2 variants of concern. In a Syrian hamster model for COVID-19, MTX reduced viral replication and inflammation in the infected lungs [106]. Two other DHFR inhibitors, pralatrexate and trimetrexate, also demonstrated effects in counteracting SARS-CoV-2 infection [107]. These studies suggest that cellular DHFR activity is required for SARS-CoV-2 infection.

### 4.2. Influenza Viruses

#### DHFR

Azaspiro dihydrotriazines and cycloguanil analogues have been reported as new inhibitors of influenza A virus (IAV) and influenza B virus (IBV) by targeting the cellular DHFR enzyme. These studies suggest that cellular DHFR activity is essential for the infections caused by influenza viruses [108,109].

### 4.3. Zika Virus

#### 4.3.1. DHFR

MTX was found to inhibit Zika virus replication in Vero cells and human neural stem cells. The addition of leucovorin, a downstream metabolite in the cellular DHFR pathway, restored Zika virus replication impaired by MTX treatment, indicating that the antiviral effect of MTX is due to inhibition of cellular DHFR. Additionally, the addition of adenosine was able to rescue the Zika virus inhibited by MTX, suggesting that the restriction of de novo synthesis of adenosine triphosphate (ATP) pools suppresses viral replication [110]. These results suggest that cellular DHFR activity is essential for Zika virus infections.

#### 4.3.2. SAMHD1

The cellular SAMHD1 was significantly upregulated in human skin fibroblasts upon infection with the Zika virus. Overexpression of SAMHD1 in cutaneous cells or pretreatment of cells with virus-like particles containing the SAMHD1 restriction factor Vpx resulted in a strong increase or inhibition, respectively, of Zika virus replication. Moreover, silencing SAMHD1 with siRNA targeting SAMHD1 led to a marked decrease in viral RNA levels. The proviral role of SAMHD1 in Zika virus infection of human skin cells is likely due to its dNTPase activity, which reduces the level of dNTPs, thus potentially increasing the NTP pool for RNA viruses. Alternatively, SAMHD1 may enhance Zika virus replication by suppressing innate immune responses, such as the interferon response [111].

### 4.4. HCV

#### RNRs

Kitab et al. demonstrated that cellular RRM2 expression is upregulated by HCV infection in quiescent hepatocytes. RRM2 is a cellular factor found to be essential for HCV replication. Mechanistically, RRM2 promotes HCV RNA replication by protecting the NS5B protein from hPLIC1-dependent proteasomal degradation. Increased RRM2 mRNA and protein expression levels were detected in HCV-infected hepatocytes from chimeric mice and hepatoma cells infected with the HCV strain JFH1 [112]. In another study, Yang et al. found that the expression of cellular RRMs, including RRM1 and RRM2, was downregulated in HCV-infected Huh7.5 cells and Huh7 cells with HCV subgenomic RNAs (HCVr). As expected, the NTP/dNTP ratio was elevated in HCVr cells. Compared to control Huh7 cells with sh-scramble, the NTP/dNTP ratio in RRM-knockdown cells was elevated. Knockdown of cellular RRM1 or RRM2 increased HCV replication in HCVr cells. Additionally, inhibitors of RRMs, such as Didox, Trimidox, and hydroxyurea, enhanced HCV replication [113].

These findings may not be contradictory; they may reflect a mechanism that allows HCV to adapt to different conditions. In quiescent hepatocytes, where intracellular NTP levels (or the NTP/dNTP ratio) are high due to no cellular DNA replication, when NTP/dNTP ratio is high enough for HCV replication (i.e., expression of RRM2 is low), RRM2 would be upregulated by HCV infection and could promote HCV replication through protecting the NS5B protein from degradation. Conversely, when NTP/dNTP ratio is low in the hepatoma cell lines (e.g., Huh7, expression of RRM2 is elevated), HCV would suppress the expression of RRMs to increase the cellular NTP/dNTP ratio to support HCV replication.

### 4.5. Chikungunya Virus

#### SAMHD1

Similar to Zika virus, SAMHD1 enhances Chikungunya virus replication, though the mechanism is not yet clear [111].

### 4.6. Respiratory Syncytial Virus

#### DHFR

Azaspiro dihydrotriazines and cycloguanil analogues have been reported as respiratory syncytial virus (RSV) inhibitors targeting the host factor DHFR, suggesting that cellular DHFR activity is essential for the infections caused by RSV [108,109].

## 5. Regulation of Viral Infections by Genes Involved in dNTP Metabolism Independent from dNTP Level

Genes involved in dNTP metabolism, such as viral RNR and cellular SAMHD1, have various functions beyond their roles in dNTP metabolism. Besides reducing NTPs to the corresponding dNTPs, viral RNRs from herpesviruses may aid in viral immune evasion by affecting protein aggregation or antagonizing the APOBEC3 family [114]. Similarly, cellular SAMHD1, in addition to its dNTPase activity, can prevent the induction of type I interferons by degrading nascent DNA at stalled replication forks [115]. SAMHD1 also enhances immunoglobulin hypermutation in B-lymphocytes [116]. Thus, both viral RNR and cellular SAMHD1 can regulate the replication of various viruses through mechanisms not directly related to dNTP metabolism.

### 5.1. RNR

The RRM1 subunit of HSV-2 RNR contains a domain similar to the alpha-crystallin domain found in small heat shock proteins. This domain is crucial for oligomerization and cytoprotective activities, which are not observed in mammalian RRM1. HSV-2 RRM1 exhibits chaperone activity, protecting cells against apoptosis, which might play a significant role in viral pathogenesis [117].

The cellular apolipoprotein B mRNA editing enzyme catalytic peptide-like protein (APOBEC) family, particularly the APOBEC3 cytosine deaminases, has been a focus of virology research for its role in inactivating and mutating several human viruses. APOBEC3B, in particular, has been recognized as a DNA editor that limits viral replication and transcription. Viruses have evolved mechanisms to counteract host immunity, such as the EBV BORF2 (viral RNR), which directly inhibits and relocates cellular APOBEC3B from the nucleus to the cytoplasm [118,119,120,121]. Similarly, the HSV1 UL39 gene product (viral RNR) appears to relocate cellular APOBEC3A from the nucleus to the cytoplasm [114,122,123]. Interestingly, HCMV is reported recently to mediate APOBEC3B relocalization through a viral RNR-independent mechanism [124].

In Avian Leukemia Virus subgroup J (ALV-J), cellular RRM2 upregulation facilitates viral replication by interacting with the viral capsid protein P27 and activating the Wnt/β-catenin signaling pathway. This interaction underscores a non-canonical role of cellular RNR in viral replication independent of nucleotide synthesis [125].

### 5.2. SAMHD1

The cellular SAMHD1 negatively regulates interferon signaling, which is a crucial component of the innate immune response. For instance, the genetic loss of SAMHD1 leads to heightened interferon activation, suppressing SARS-CoV-2 replication, indicating its role in modulating immune responses during viral infection [126].

SAMHD1 has been shown to inhibit Human T Cell Leukemia Virus 1 (HTLV-1) infection in macrophages by activating a STING-mediated apoptosis pathway, highlighting its role in antiviral immune responses [127].

The chemokine CCL5 upregulates cellular SAMHD1, which then inhibits the replication of Influenza A Virus (IAV) in alveolar epithelial cells. Knockdown of SAMHD1 resulted in enhanced IAV replication and abolished the CCL5-mediated inhibition of IAV replication in A549 cells [128]. Another report also suggested that cellular SAMHD1 acts as a host restriction factor against IAVs, which may be countered by the viral PA protein [129].

The cellular SAMHD1 restricts multiple enteroviruses, such as enterovirus 71 (EV71) and enterovirus D68 (EVD68), by interfering with viral assembly. It achieves this by binding to the VP1 domain, which is crucial for the interaction with VP2, thereby inhibiting the formation of viral particles. This study reveals a novel mechanism for SAMHD1’s anti-EV activity [130].

There is some controversy regarding the cellular SAMHD1’s ribonuclease (RNase) activity, with conflicting reports from different research groups [131,132,133,134]. Further studies are needed to clarify whether the cellular SAMHD1 possesses this activity.

Fish SAMHD1, similar to those in mammals, acts as an innate immunity restriction factor. On the other hand, it also inhibits viral infection through IRF3-mediated antiviral and apoptotic responses. For example, SAMHD1 from grass carp can suppress the proliferation of grass carp reovirus, although the exact mechanism remains unclear [135].

Overall, viral RNR and cellular SAMHD1 play multifaceted roles in regulating viral infections. They not only influence nucleotide metabolism but also engage in diverse activities, which highlight the complexity of host–virus interactions.

## 6. Application of Oncolytic Viruses without Viral RNRs

Completion of viral life cycle within the cells could lyse the viral infected cells. Thus, viruses could be oncolytic, i.e., to lyse the infected cancer cells. Cancer cells often have elevated levels of dNTPs due to their rapid proliferation [4], which is not a characteristic feature of normal cells. Oncolytical viruses (wild-type or genetically modified) could exploit this difference to target and replicate selectively within cancer cells, sparing normal, healthy cells [136]. Indeed, Myxoma virus, which does not have viral RRM1 gene, is an oncolytic poxvirus against soft tissue sarcomas with higher cellular RRM2 expression [137]. Furthermore, a novel oncolytic VV has been developed by mutating the F4L gene, which encodes viral RRM2. The F4L-deleted VVs are highly attenuated in normal tissues, but since cancer cells often express elevated RRM2 levels and have elevated levels of dNTPs, these viruses exhibit tumor-selective replication and then cell killing [138,139]. Similarly, the UL39 gene of human HSV-1 encodes viral RRM1, also known as ICP6, to elevate the cellular dNTPs level for viral replication. Targeting UL39 with CRISPR-Cas9 could potentially develop an oncolytic HSV-1 virus [140].

Cellular DHFR activity is required for cancer cells [141] and the cellular SAMHD1 protein is a tumor suppressor whose expression is reduced in cancer cells [142]. As expected, the cellular DHFR inhibitors are potential anti-cancer agents [141,143]. Neither VV nor HSV-1 encodes a viral DHFR. The combination of these oncolytic viruses and the cellular DHFR inhibitors should exhibit better tumor-cell-specific killing.

## 7. Conclusions

The cellular enzymes involved in dNTP metabolism, including DHFR, RNR, and SAMHD1, play crucial roles in viral replication and host–pathogen interactions. Viruses have evolved various mechanisms to manipulate these cellular enzymes to create a favorable environment for their replication (Table 1). Several common themes that emerge from these host cell–virus interactions could be found in this review: (1) the cellular DHFR enzyme activity is essential for the biosynthesis of purines, thymidine, and glycine, making it crucial for the replication of all viruses. Some DNA viruses encode their own DHFR, while others upregulate cellular DHFR expression (Table 1); (2) the cellular RNR activity increases the dNTP/NTP ratio, benefiting the replication of DNA viruses, retroviruses, and HBV by enhancing dNTP availability. Some DNA viruses encode their own RNR, while others upregulate cellular RNR expression (Table 1); (3) the dNTPase activity of the cellular SAMHD1 decreases the dNTP/NTP ratio, thereby restricting the replication of DNA viruses, retroviruses, and HBV. To overcome this restriction, these viruses have evolved various mechanisms to suppress SAMHD1 activity (Table 1). The interactions between large DNA viruses (e.g., herpesviruses) and the genes involved in dNTP metabolism have been extensive studied. To unveil the connections between viral infections and genes involved in dNTP metabolism completely, there are still the gaps in the current knowledge regarding the interactions of these genes with small DNA viruses (e.g., parvoviruses), and especially with various RNA viruses.

In addition to cellular DHFR, RNR, and SAMHD1, thymidine kinase (TK) genes are also well-studied in the context of dNTP metabolism. For example, alpha-herpesviruses can encode their own TK genes to benefit viral replication [144]. While much of the research mentioned in this review has focused on individual genes involved in dNTP metabolism, future research directions may need to study the simultaneous regulation of multiple genes in this pathway during viral infections. For instance, studies have shown that the constitutively active retinoblastoma tumor suppressor can attenuate the expression of specific dNTP synthetic cellular enzymes, including DHFR, RRM1, RRM2, and thymidylate synthase (TS) [145]. Therefore, comprehensive transcriptomic and proteomic analyses are necessary to fully understand the relationship between dNTP biosynthesis and viral infections, providing insights into potential anti-viral therapeutic targets and strategies. Moreover, determination of NTP and dNTP molecules should be conducted to clarify the interactions of viral infections with genes involved in the dNTP metabolism due to the fact that these genes may have other functions not involved in dNTP metabolism, such as viral RNRs and cellular SAMHD1.

## Figures and Tables

**Figure 1 viruses-16-01412-f001:**
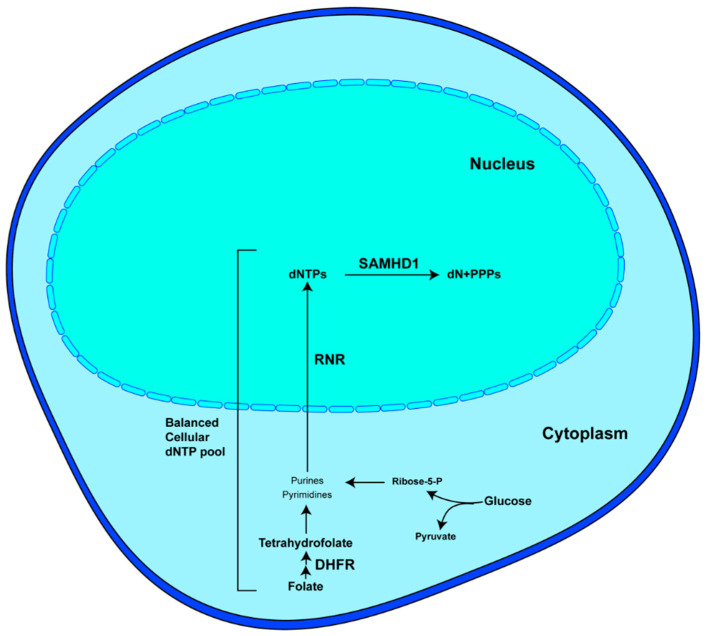
Metabolism of dNTPs in normal cells [1,2,3,4]. Nucleotides could be derived from multiple intracellular sources, such as the products of glycolysis and folate cycle. Dihydrofolate is reduced to active tetrahydrofolate by dihydrofolate reductase (DHFR). Ribonucleotide reductase (RNR) reduces both pyrimidine and purine bases to deoxynucleosides. SAM and HD domain-containing protein 1 (SAMHD1) hydrolyzes dNTPs into deoxynucleosides (dNs) and inorganic triphosphates (PPPi).

**Figure 2 viruses-16-01412-f002:**
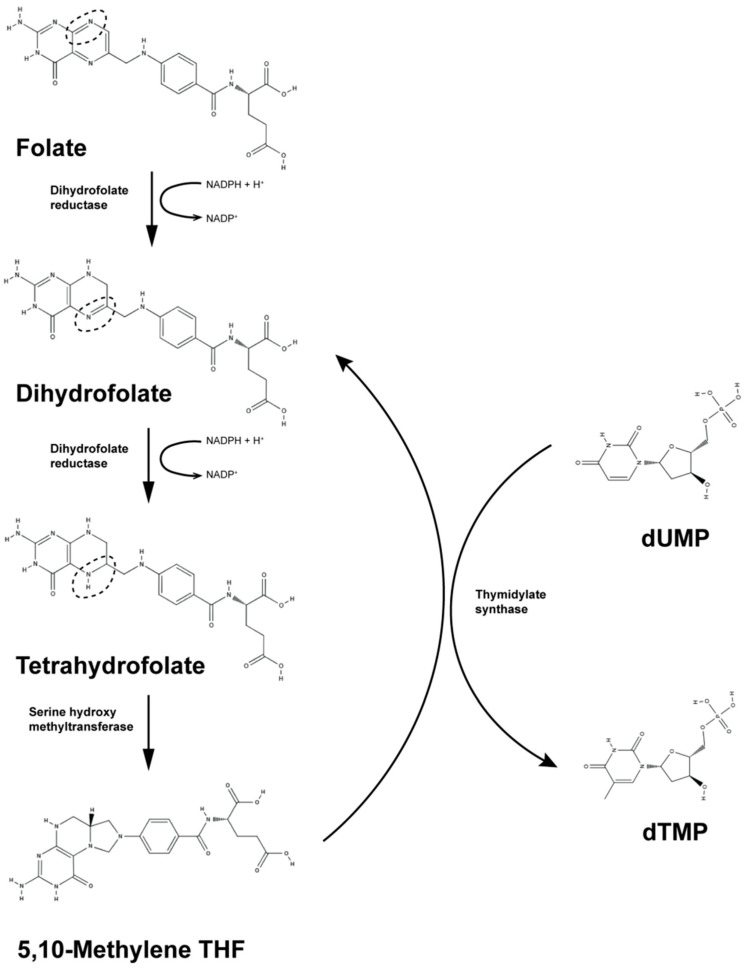
Folate metabolic pathway [5,6,7,8]. DHFR catalyzes the transfer of a hydride from the cofactor nicotinamide adenine dinucleotide phosphate (NADPH), which serves as an electron donor, to dihydrofolate (DHF), resulting in the production of tetrahydrofolate (THF) through protonation. Specifically, DHFR facilitates the reduction of 7,8-dihydrofolate to 5,6,7,8-tetrahydrofolate using NADPH as a cofactor. Additionally, DHFR works in conjunction with thymidylate synthase, which catalyzes the reductive methylation of deoxyuridine monophosphate (dUMP) to deoxythymidine monophosphate, using N5-N10-methylenetetrahydrofolate (5,10-Methylene THF) as a cofactor.

**Figure 3 viruses-16-01412-f003:**
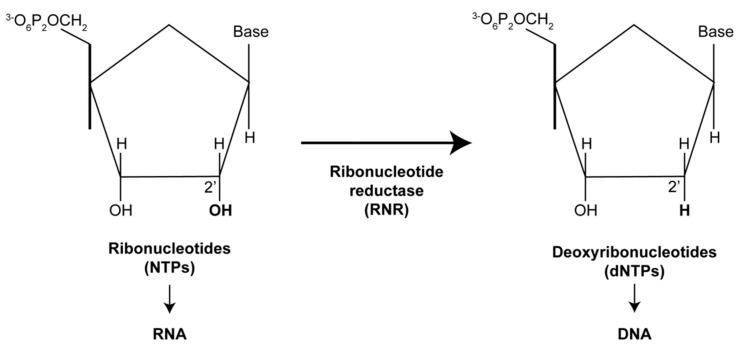
Ribonucleotide reductase (RNR) converts the building blocks of RNA into those of DNA. In mammals, RNR enzymes reduce the 2′ carbon of nucleoside diphosphates (NDPs) to form the corresponding deoxynucleoside diphosphates (dNDPs). These dNDPs are then phosphorylated by nucleoside diphosphate kinase to produce deoxynucleoside triphosphates (dNTPs), which are essential for nuclear and mitochondrial DNA replication and repair [9].

**Figure 4 viruses-16-01412-f004:**
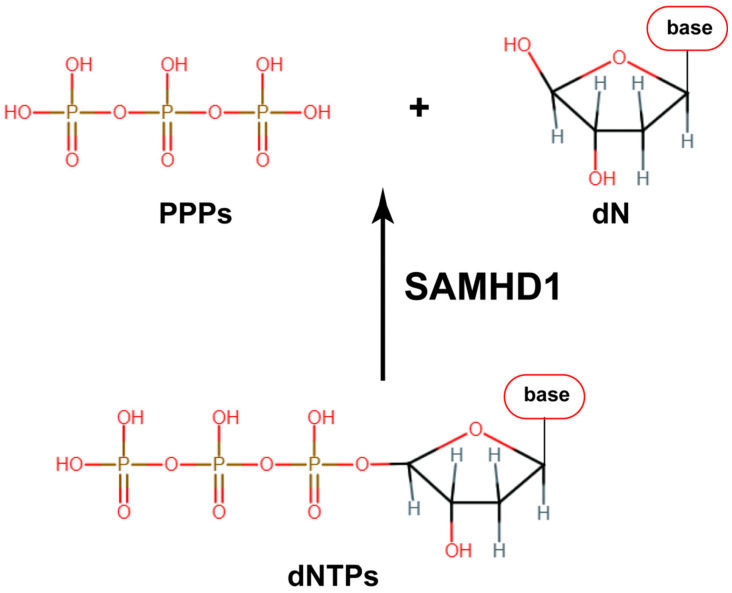
SAMHD1, a deoxyribonucleoside triphosphate triphosphohydrolase, hydrolyzes dNTPs into deoxynucleosides (dNs) and inorganic triphosphates (PPPi) that are either recycled or degraded, thereby limiting the dNTP pool and impairing DNA replication [14].

**Table 1 viruses-16-01412-t001:** Expression of genes involved in dNTP metabolism modulated by various human viruses.

Virus	Genes	References
HSV	HSVs encode viral RRM1 and RRM2.	[24,25]
	HSVs encode a viral protein kinase to phosphorylate cellular SAMHD1 protein and inhibit its dNTPase activity.	[43,44]
VZV	VZV encodes viral RRM1 and RRM2.	[32]
	VZV encodes a viral protein kinase to phosphorylate cellular SAMHD1 protein and inhibit its dNTPase activity.	[43,44]
EBV	EBV encodes viral RRM1 and RRM2.	[36,37]
	EBV encodes a viral protein kinase to phosphorylate cellular SAMHD1 protein and inhibit its dNTPase activity.	[43,44]
CMV	CMV activates the cellular DHFR gene expression.	[17,21]
	CMV encodes a viral protein kinase to phosphorylate cellular SAMHD1 protein and inhibit its dNTPase activity; CMV could also downregulate or relocalize cellular SAMHD1.	[43,44]
HHV-6/7	HHV-6/7 encode a viral protein kinase to phosphorylate cellular SAMHD1 protein and inhibit its dNTPase activity.	[43,44]
HHV-8	HHV-8 contains a viral DHFR gene.	[18]
	HHV-8 encodes viral RRM1 and RRM2.	[36,37]
	HHV-8 encodes a viral protein kinase to phosphorylate cellular SAMHD1 protein and inhibit its dNTPase activity.	[43,44]
Vaccinia virus (VV)	VV encodes viral RRM1 and RRM2.	[49]
	Cellular SAMHD1 protein significantly inhibits VV infection.	[39]
Adenovirus	Adenovirus activates the cellular DHFR gene expression.	[59,60]
HPV	HPV31 upregulates cellular RRM2 expression.	[62]
	HPV16 replication is enhanced in the absence of cellular SAMHD1 protein.	[65]
HIV	Reduction of RRM2 expression suppresses HIV-1 replication.	[68]
	The cellular SAMHD1 protein significantly inhibits HIV-1 infection.	[15]
	HIV-2 and several SIV strains antagonize the cellular SAMHD1 restriction using Vpx.	[16,84,86]
HBV	HBV X gene upregulates cellular RRM2 expression.	
	Cellular SAMHD1 protein inhibits HBV infection.	[93,94,95,96]
SARS-CoV-2	The cellular DHFR activity is required for SARS-CoV-2 infection.	[106,107]
Influenza Viruses	The cellular DHFR activity is essential for the infections of influenza viruses	[108,109]
Zika Virus	The cellular DHFR activity is essential for Zika virus infections.	[110]
	Zika virus upregulates the cellular SAMHD1 expression.	[111]
HCV	The cellular RRM2 expression is upregulated by HCV infection in quiescent hepatocytes.	[112]
	The expression of cellular RRMs was downregulated in HCV-infected Huh7.5 cells.	[113]

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
