# Peer review of "Unveiling the Connection: Viral Infections and Genes in dNTP Metabolism"

_viruses, 2024, doi:10.3390/v16091412_

Round 1

Reviewer 1 Report

Comments and Suggestions for Authors

Lo and colleagues summarize the role of dNTP metabolisms in viral replication. The review covers various viruses including RNA viruses. It is an interesting review and concisely summarizes findings in the field. The review could be enriched if the authors could add more descriptions, summary sentences, and discussions, since it is very descriptive currently. Other points are below.

-       Descriptions on SAMHD1 need more clarifications. For example, in Line 161, the authors state “SAMHD1 is regulated by cyclin-dependent kinases”. Which aspect of SAMHD1 is regulated? Expression? dNTPase activity, or Antiviral activity? Please clarify. Similarly, Line 162, “CDK6 and CDK2, mediate SAMHD1 activation” is ambiguous.

-       Line 165: Do viral CDK-like proteins other than EBV BGLF4 regulate SAMHD1 activity?

-       Line 166: Are those evasion strategies for which viruses?

-       Line 228: It seems this sentence is a copy of the original paper’s title, but the meanings is unclear. Please clarify their findings.

-       Section 3 should be either divided into 2 sections or rephrased. There is no “retrovirus-like” virus; HIV is a retrovirus. If the authors discuss retroviruses and HBV  in the same section since they use RT during their replications, that fact should be stated in the paragraph but not “retrovirus-like” virus.

-       Line 254. Typo.

-       Line 263: First-choice ART regimen is two NRTIs, and PI, NNRTI or INI.

-       Line 274: Please use “person-first languages”. Details can be found, e.g. at:

https://www.nih.gov/nih-style-guide/person-first-destigmatizing-language

-       Line: 281: Some SIVs use Vpr to antagonize SAMHD1.

-       Line 283: Is “usingand” a typo?

-       Line 283: How do HIV-2 and SIVs “elevate intracellular dNTP pools”?

-       Line 390: The sentence seems incomplete.

-       Section 5: Is there any discussion on the role of DHFR in viral replication other than its activity on folate metabolisms? Just curious.

-       The objective of Section 6 is unclear. If the authors would like to discuss the usage of  mutant viruses such as UL39-null HSV-1 as a tool to selectively kill tumor cells, the section requires more descriptions on backgrounds, current research, future directions etc.

Comments on the Quality of English Language

There are some typos, incomplete sentences etc. 

Author Response

Reviewer 1

Lo and colleagues summarize the role of dNTP metabolisms in viral replication. The review covers various viruses including RNA viruses. It is an interesting review and concisely summarizes findings in the field. The review could be enriched if the authors could add more descriptions, summary sentences, and discussions, since it is very descriptive currently. Other points are below.

-       Descriptions on SAMHD1 need more clarifications. For example, in Line 161, the authors state “SAMHD1 is regulated by cyclin-dependent kinases”. Which aspect of SAMHD1 is regulated? Expression? dNTPase activity, or Antiviral activity? Please clarify. Similarly, Line 162, “CDK6 and CDK2, mediate SAMHD1 activation” is ambiguous.

Response: Thanks for the suggestion. We have clarified the statements in lines 168-170 of the revised manuscript labeled with red color.

-       Line 165: Do viral CDK-like proteins other than EBV BGLF4 regulate SAMHD1 activity?

Response: Thanks for the question. We have added the information in lines 171-174 of the revised manuscript.

-       Line 166: Are those evasion strategies for which viruses?

Response: Thanks for the question. We have clarified the statements in line 175 of the revised manuscript.

-       Line 228: It seems this sentence is a copy of the original paper’s title, but the meanings is unclear. Please clarify their findings.

Response: Thanks for the suggestion. We have clarified the statements in lines 234-235 of the revised manuscript.

-       Section 3 should be either divided into 2 sections or rephrased. There is no “retrovirus-like” virus; HIV is a retrovirus. If the authors discuss retroviruses and HBV  in the same section since they use RT during their replications, that fact should be stated in the paragraph but not “retrovirus-like” virus.

Response: Thanks for the correction. We have revised the term in line 250 of the revised manuscript.

-       Line 254. Typo.

Response: Thanks for the correction. We have corrected the mistake.

-       Line 263: First-choice ART regimen is two NRTIs, and PI, NNRTI or INI.

Response: Thanks for the correction. We have added the information in lines 272-274 of the revised manuscript.

-       Line 274: Please use “person-first languages”. Details can be found, e.g. at:

https://www.nih.gov/nih-style-guide/person-first-destigmatizing-language

Response: Thanks for the correction. We have corrected the sentence in line 284 of the revised manuscript.

-       Line: 281: Some SIVs use Vpr to antagonize SAMHD1.

Response: Thanks for the correction. We have added the information in lines 295-296 of the revised manuscript.

-       Line 283: Is “usingand” a typo?

Response: Thanks for the correction. We have re-written the sentence in line 294 of the revised manuscript.

-       Line 283: How do HIV-2 and SIVs “elevate intracellular dNTP pools”?

Response: Thanks for the suggestion. We have added the information in lines 293-295 of the revised manuscript.

-       Line 390: The sentence seems incomplete.

Response: Thanks for the correction. We have re-written the sentence in lines 411-412 of the revised manuscript.

-       Section 5: Is there any discussion on the role of DHFR in viral replication other than its activity on folate metabolisms? Just curious.

Response: Thanks for the question. This is beyond my knowledge. I could not discuss it.

-       The objective of Section 6 is unclear. If the authors would like to discuss the usage of mutant viruses such as UL39-null HSV-1 as a tool to selectively kill tumor cells, the section requires more descriptions on backgrounds, current research, future directions etc.

Response: Thanks for the suggestion. We have re-written the Section 6 in lines 487-501 of the revised manuscript.

Comments on the Quality of English Language

There are some typos, incomplete sentences etc.

Response: Thanks for the correction. We have checked them.

Reviewer 2 Report

Comments and Suggestions for Authors

This manuscript, submitted as a review article by Lo and colleagues, summarizes the regulation of genes involved in dNTP metabolism and their interaction with different viruses. The authors briefly introduce dNTP metabolism in normal cells and discuss the roles of key enzymes, including dihydrofolate reductase (DHFR), ribonucleotide reductase (RNR), and SAM domain and HD domain-containing protein 1 (SAMHD1). They then describe the status of each of these enzymes in relation to different viruses. This is an interesting attempt to summarize how these enzymes affect dNTP concentration during viral infection, which could lead to potential therapeutic avenues and provide a general understanding of virus-host interaction pathways. However, this review merely summarizes published results rather than emphasizing the significance of understanding the connection between viruses and genes involved in dNTP metabolism. Several key points are missing in this review, which are summarized below.

Major points:

  1. The abstract states, “This review provides a concise summary of the interactions between different viruses and the genes involved in dNTP metabolism”; however, the authors do not clarify whether this review focuses on human genes, viral genes, or both. At some points, they mention viral-encoded genes, but it is often unclear whether certain viruses encode these genes or not. Additionally, the authors do not specify whether human dNTP genes play a role against all viruses or only certain types.

  2. I strongly encourage the authors to state the genome architecture of DNA and RNA viruses, including whether they are double or single-stranded, segmented versus non-segmented, and positive versus negative sense for each of the viruses mentioned in this review.

  3. A summary table containing information corresponding to each virus, the affected genes, and whether these genes are of human or viral origin would help readers follow the theme.

  4. The mechanism by which certain viruses modulate dNTP genes should be addressed, rather than merely stating that they affect virus replication. Similarly, the mechanism by which different inhibitors affect viral replication should also be discussed.

  5. In the conclusion section, it would be appropriate to mention the gaps in current knowledge, suggest future research directions, and highlight any common themes that emerge from the interactions between different viruses and dNTP genes.

Comments on the Quality of English Language

Minor editing required

Author Response

Reviewer 2

This manuscript, submitted as a review article by Lo and colleagues, summarizes the regulation of genes involved in dNTP metabolism and their interaction with different viruses. The authors briefly introduce dNTP metabolism in normal cells and discuss the roles of key enzymes, including dihydrofolate reductase (DHFR), ribonucleotide reductase (RNR), and SAM domain and HD domain-containing protein 1 (SAMHD1). They then describe the status of each of these enzymes in relation to different viruses. This is an interesting attempt to summarize how these enzymes affect dNTP concentration during viral infection, which could lead to potential therapeutic avenues and provide a general understanding of virus-host interaction pathways. However, this review merely summarizes published results rather than emphasizing the significance of understanding the connection between viruses and genes involved in dNTP metabolism. Several key points are missing in this review, which are summarized below.

Major points:

The abstract states, “This review provides a concise summary of the interactions between different viruses and the genes involved in dNTP metabolism”; however, the authors do not clarify whether this review focuses on human genes, viral genes, or both. At some points, they mention viral-encoded genes, but it is often unclear whether certain viruses encode these genes or not. Additionally, the authors do not specify whether human dNTP genes play a role against all viruses or only certain types.

Response: Thanks for the suggestion. To specify this, we added [cellular] and [viral] before the genes in the revised manuscript labeled with red color.

I strongly encourage the authors to state the genome architecture of DNA and RNA viruses, including whether they are double or single-stranded, segmented versus non-segmented, and positive versus negative sense for each of the viruses mentioned in this review.

Response: Thanks for the suggestion. We have added the information in lines 119-122, 252-253, 348-352 of the revised manuscript.

A summary table containing information corresponding to each virus, the affected genes, and whether these genes are of human or viral origin would help readers follow the theme.

Response: Thanks for the suggestion. We have added the information as suggested in Table 1 of the revised manuscript.

The mechanism by which certain viruses modulate dNTP genes should be addressed, rather than merely stating that they affect virus replication. Similarly, the mechanism by which different inhibitors affect viral replication should also be discussed.

Response: Thanks for the comment. The mechanisms regarding certain viruses modulate dNTP genes were written in sections 2-1-1, 2-1-2, 2-1-3, 2-2-2, 2-3-1, 2-4-2, 3-1-3, 3-2-2, and 4-2-2. If we did not write the mechanisms regarding different inhibitors affect viral replication, that is due to the mechanisms are not understood.

In the conclusion section, it would be appropriate to mention the gaps in current knowledge, suggest future research directions, and highlight any common themes that emerge from the interactions between different viruses and dNTP genes.

Response: Thanks for the suggestion. We have tried to add more information in lines 524-527 of the revised manuscript.

Comments on the Quality of English Language

Minor editing required

Response: Thanks for the suggestion. We have tried hard to correct them.

Round 2

Reviewer 1 Report

Comments and Suggestions for Authors

The authors responded well to the critiques, and most of the concerns were addressed. A few remained concerns are:

1. The previous concern, “Line 228: It seems this sentence is a copy of the original paper’s title, but the meaning is unclear. Please clarify their findings.”, was not addressed. The new line number is 232. Does “recruitment of SAMHD1 to replicating viral DNA” have a pro-viral role? If so, please state it.

2. Line 247 is inaccurate for retroviruses, and the sentence applies only for HBV. Retroviral genomes are RNA, they don’t use RT to replicate DNA, and retroviral DNA (but not RNA) is an intermediate form.

Author Response

The authors responded well to the critiques, and most of the concerns were addressed. A few remained concerns are:

  1. The previous concern, “Line 228: It seems this sentence is a copy of the original paper’s title, but the meaning is unclear. Please clarify their findings.”, was not addressed. The new line number is 232. Does “recruitment of SAMHD1 to replicating viral DNA” have a pro-viral role? If so, please state it.

Response: Thanks for the comment. Sorry for the mis-understanding in the previous revision. Yes! It has a pro-viral role, which is added in the lines 232-233 of the revised manuscript labeled in red.

  1. Line 247 is inaccurate for retroviruses, and the sentence applies only for HBV. Retroviral genomes are RNA, they don’t use RT to replicate DNA, and retroviral DNA (but not RNA) is an intermediate form.

Response: We appreciate your correction! These sentences have been corrected in the lines 248-250 of the revised manuscript labeled in red.

Reviewer 2 Report

Comments and Suggestions for Authors

The authors have addressed some comments, but they still have not added the importance of this review. They mentioned, “We have tried to add more information in lines 524-527 of the revised manuscript,” in the conclusion section; however, there is no such information provided in the revised manuscript. I recommend mentioning the gaps in current knowledge, suggesting future research directions, and highlighting any common themes that emerge from the interactions between different viruses and dNTP genes in the conclusion section.

Comments on the Quality of English Language

The Quality of English language is fine

Author Response

The authors have addressed some comments, but they still have not added the importance of this review. They mentioned, “We have tried to add more information in lines 524-527 of the revised manuscript,” in the conclusion section; however, there is no such information provided in the revised manuscript. I recommend mentioning the gaps in current knowledge, suggesting future research directions, and highlighting any common themes that emerge from the interactions between different viruses and dNTP genes in the conclusion section.

Response: Thanks for the suggestions. [the gaps in current knowledge, future research directions, and common themes that emerge from the interactions between different viruses and dNTP genes] were mentioned in the conclusion section as suggested in the lines 497-519 of the revised manuscript labeled in red.
